# Protective Effects of a Jellyfish-Derived Thioredoxin Fused with Cell-Penetrating Peptide TAT-PTD on H_2_O_2_-Induced Oxidative Damage

**DOI:** 10.3390/ijms24087340

**Published:** 2023-04-16

**Authors:** Bo Wang, Peipei Zhang, Qianqian Wang, Shuaijun Zou, Juxingsi Song, Fuhai Zhang, Guoyan Liu, Liming Zhang

**Affiliations:** 1Department of Marine Biomedicine and Polar Medicine, Naval Special Medical Center, Naval Medical University, Shanghai 200433, China; 2Department of Infectious Disease, No. 971 Hospital of the PLA Navy, Qingdao 266071, China; 3Department of Marine Biological Injury and Dermatology, Naval Special Medical Center, Naval Medical University, Shanghai 200052, China

**Keywords:** thioredoxin, oxidative stress, jellyfish, protein transduction domain, antioxidant

## Abstract

Thioredoxin (Trx) plays a critical role in maintaining redox balance in various cells and exhibits anti-oxidative, anti-apoptotic, and anti-inflammatory effects. However, whether exogenous Trx can inhibit intracellular oxidative damage has not been investigated. In previous study, we have identified a novel Trx from the jellyfish *Cyanea capillata*, named CcTrx1, and confirmed its antioxidant activities in vitro. Here, we obtained a recombinant protein, PTD-CcTrx1, which is a fusion of CcTrx1 and protein transduction domain (PTD) of HIV TAT protein. The transmembrane ability and antioxidant activities of PTD-CcTrx1, and its protective effects against H_2_O_2_-induced oxidative damage in HaCaT cells were also detected. Our results revealed that PTD-CcTrx1 exhibited specific transmembrane ability and antioxidant activities, and it could significantly attenuate the intracellular oxidative stress, inhibit H_2_O_2_-induced apoptosis, and protect HaCaT cells from oxidative damage. The present study provides critical evidence for application of PTD-CcTrx1 as a novel antioxidant to treat skin oxidative damage in the future.

## 1. Introduction

Skin is one of the organs most vulnerable to various environmental harmful factors, including UV radiation, hypothermia, particulate matter, and other air pollutants [1,2,3,4]. The damage to skin is mainly related to oxidative stress induced by the accumulation of reactive oxygen species (ROS) [5,6,7]. In order to reduce the damage by ROS, the body develops a complete set of antioxidant systems to maintain the balance of free radicals. Common antioxidant enzymes include superoxide dismutase (SOD), catalase (CAT), glutathione peroxidase (GSH-Px), thioredoxin (Trx), and thioredoxin peroxidase (TPx) [8,9].

Thioredoxin (Trx), composed of Trx1 (in cytosol) and Trx2 (in mitochondria), is a class of thermostable hydrogen carrier proteins distributed in almost all organisms and can be functionalized as an important hydrogen donor in many reduction reactions [10,11]. As the key regulator of intracellular redox balance, Trx plays critical roles in a series of life activities, including anti-oxidative, antitumor, and anti-apoptotic processes [12,13,14]. Moreover, Trx can defend the body against oxidative stress through its disulfide reductase activity and ROS scavenging ability and is regarded as an important source for antioxidant drug development [12,15].

As a representative class of macroplankton, jellyfish are continuously exposed to high light, UV radiation, as well as other harsh environmental factors and are believed to possess robust mechanisms to maintain the intracellular redox balance [16]. Several studies have confirmed that jellyfish contained a variety of highly active antioxidants [16,17,18,19]. In previous research, we have identified a novel Trx1, named CcTrx1 from the jellyfish *Cyanea capillata* [20]. Our results illustrated that CcTrx1 exhibited potent antioxidant activities and protected *C. capillata* against oxidative stress. CcTrx1 is a natural, safe, and exploitable antioxidant with a promising application prospect. However, it is difficult for exogenous proteins to enter cells and tissues, which greatly limits the application of CcTrx1 in drug development.

The protein transduction domain (PTD), also called cell-penetrating peptide (CPP), is a kind of short peptide containing 5–30 amino acid residues, rich in basic amino acids [21,22]. It can effectively carry various biomacromolecules into cells to exert biological effects through an endocytic mechanism or direct penetration of the cell membrane [23]. A series of CPPs have been studied for agent delivery. Among them, TAT-PTD, derived from HIV-1 TAT protein, is the first identified CPP, which has been successfully applied to drug delivery in vitro and in vivo [24,25,26,27].

In the present study, we combined TAT-PTD with CcTrx1 by molecular cloning and obtained sufficient amounts of PTD-CcTrx1 fusion protein with high purity. We further detected its antioxidant activities and transmembrane ability and explored its protective effects against oxidative damage. This study provides a scientific basis for the feasibility of marine antioxidant enzymes entering cells to play biological roles and for further exploring the enzyme application in drug development.

## 2. Results

### 2.1. Characterization and Identification of PTD-CcTrx1 Coding Sequence

The coding sequence of PTD-CcTrx1 composed of a 33 bp PTD coding region and a 312 bp open reading frame (ORF) of CcTrx1, which encoded a putative protein of 115 amino acids with a calculated molecular mass of 13.1 kDa and an estimated pI of 9.0 (Appendix A). The recombinant plasmid of PTD-CcTrx1 was double-digested by NdeⅠ and XhoⅠ for nucleic acid electrophoresis and sequencing identification. Gel electrophoresis analysis of the digestion products preliminarily indicated the accuracy of inserted sequence (Appendix A), and the sequencing result finally confirmed that there were no mutations in the inserted PTD-CcTrx1 coding sequence.

### 2.2. Expression and Purification of PTD-CcTrx1

The fusion protein PTD-CcTrx1 was expressed under optimal conditions based on the results of preliminary experiments and purified via the ÄKTA purifier system. As shown in Figure 1a, compared with the uninduced samples (lane 1), PTD-CcTrx1 was sufficiently expressed in the supernatants of sonicated lysates after induction (lane 4). As shown in Figure 1b, a protein band with very high purity was shown in the fraction of 30% eluting peak (lane 4). The actual molecular weight corresponded well with the calculated molecule weight of recombinant PTD-CcTrx1. Furthermore, the fusion protein PTD-CcTrx1 was confirmed via the Western blot analysis using an anti-His-tag antibody (Figure 1c).

### 2.3. Transmembrane Delivery of PTD-CcTrx1 into HaCaT Cells

To investigate whether the fusion protein PTD-CcTrx1 could be delivered into cells, we detected the level of intracellular His-tagged proteins via the Western blot assay and immunofluorescence assay. As shown in Figure 2a,b, PTD-CcTrx1 could efficiently enter HaCaT cells 15 min after incubation, and the amount of the intracellular PTD-CcTrx1 gradually increased in a time-dependent manner (15 min to 2 h), while the CcTrx1 could not enter the cells. The result of immunofluorescence assay (Figure 2c,d) is consistent with that of the Western blot assay, further confirming the noticeable transmembrane ability of PTD-CcTrx1. These results indicated that PTD could deliver CcTrx1 into HaCaT cells efficiently.

### 2.4. PTD-CcTrx1 Exhibited Specific Antioxidant Activities in Cell-Free System

An insulin disulfide reduction assay was performed to investigate the disulfide reductase activity of PTD-CcTrx1. As shown in Figure 3a, PTD-CcTrx1 exhibited specific activity to reduce insulin disulfides in a time-dependent manner, and the activity increased significantly with an increase in concentration. In addition, no significant difference was observed between the disulfide reductase activities of PTD-CcTrx1 and CcTrx1. These results indicated that PTD-CcTrx1 efficiently retained disulfide reductase activity.

Furthermore, the MCO assay was performed to evaluate the ability of PTD-CcTrx1 to protect supercoiled plasmid DNA from oxidative damage. As shown in Figure 3b,c, supercoiled DNA separately incubated with DTT was not damaged (lane 2), while the DNA incubated with both FeCl_3_ and DTT was apparently converted from the supercoiled form (SF) to nicked form (NF) (lane 3). The addition of PTD-CcTrx1 to the MCO system effectively decreased the NF amount of plasmid DNA in a dose-dependent manner (lanes 4–7). No significant difference was observed between the DNA-protected activity of PTD-CcTrx1 (lane 7) and CcTrx1 (lane 8). These results indicated that PTD-CcTrx1 could effectively protect DNA from cleavage caused by the MCO system.

### 2.5. PTD-CcTrx1 Antagonized H_2_O_2_-Induced Cytotoxicity

To detect the protective effect of PTD-CcTrx1 against oxidative stress-induced cytotoxicity, an experimental model of HaCaT cells oxidatively damaged by H_2_O_2_ was established. As shown in Figure 4a,b, cell viability in the presence of H_2_O_2_ was progressively reduced in a dose-dependent manner, while both PTD-CcTrx1 and CcTrx1 exerted no obvious cytotoxicity on HaCaT cells (Figure 4c). H_2_O_2_-induced cytotoxicity was significantly inhibited by 10 μM PTD-CcTrx1, while the equal dose of CcTrx1 exhibited no inhibitory effects (Figure 4d). Moreover, pre- and posttreatment of PTD-CcTrx1 exhibited consistent protective effects towards H_2_O_2_-induced cytotoxicity in HaCaT cells.

### 2.6. PTD-CcTrx1 Protected HaCaT Cells from H_2_O_2_-Induced Apoptosis

Excessive oxidative stress mediated by H_2_O_2_ can induce cell apoptosis. To further investigate the protective effects of PTD-CcTrx1 toward H_2_O_2_-induced apoptosis and necrosis, the apoptosis assay was conducted and LDH activity was detected. As shown in Figure 5a,b, H_2_O_2_-induced apoptosis of HaCaT cells could be significantly inhibited by PTD-CcTrx1, but not by CcTrx1. The attenuation of LDH activity by PTD-CcTrx1 (Figure 5c) further confirmed the protective effect of PTD-CcTrx1 against H_2_O_2_-induced cell impairment.

### 2.7. PTD-CcTrx1 Inhibited H_2_O_2_-Mediated Oxidative Stress in HaCaT Cells

To explore the effect of PTD-CcTrx1 on the intracellular oxidative stress mediated by H_2_O_2_, a series of oxidative stress-related indicators including T-AOC, ROS, and MDA was detected. The results showed that PTD-CcTrx1 treatment could significantly increase the level of intracellular T-AOC in a dose-dependent manner (Figure 6a). H_2_O_2_ almost reduced the T-AOC by 75%, which could be largely restored by 10 μM PTD-CcTrx1, but not by CcTrx1 (Figure 6b).

ROS is a major factor in cell damage caused by oxidative stress [28,29]. As shown in Figure 6c, H_2_O_2_ could induce the accumulation of ROS in cells, which was significantly alleviated by PTD-CcTrx1. Meanwhile, the CcTrx1 exhibited no similar mitigative effect.

The major damage caused by ROS, especially by H_2_O_2_, is the oxidation of proteins and lipids [29]. We further detected the intracellular MDA to evaluate the level of lipid peroxidation. As shown in Figure 6d, intracellular MDA increased to nearly 3-fold of the control group value after exposure to H_2_O_2_. Pre- and posttreatment of PTD-CcTrx1 could reduce the MDA level by 51% and 46%, respectively, while CcTrx1 exhibited no significant mitigative effect.

### 2.8. PTD-CcTrx1 Inhibited the Activation of Apoptosis Signaling Pathways Induced by H_2_O_2_

To observe the intracellular changes and explore the underlying mechanisms of PTD-CcTrx1 against H_2_O_2_-induced apoptosis, we further detected the expression of Bcl-2, Bax, cleaved caspase 9, cleaved caspase 3, and cytochrome c (in mitochondria) via the Western blot assay. As shown in Figure 7a,b, compared with the control group, H_2_O_2_ significantly elevated the levels of Bax, cleaved caspase 9, and cleaved caspase 3 in cytoplasm, and reduced the levels of Bcl-2 in cytoplasm and cytochrome c in mitochondria, while PTD-CcTrx1 significantly attenuated the changes in expression levels of these apoptotic pathway-associated molecules, exhibiting a specific inhibitory effect on H_2_O_2_-induced mitochondrial apoptosis pathway.

## 3. Discussion

Accumulating evidence revealed that skin damage, especially chronic injuries, is mostly related to oxidative stress [7,30]. Therefore, antioxidants possess great potential to protect skin cells against oxidative damage and act as an important source for developing novel skin-defense drugs.

Trx, functionalized as a protein-disulfide reductase, is an essential component of the intracellular antioxidant system and plays a pivotal role in regulating multiple cellular redox signaling pathways [31]. Trx has shown a variety of biological functions, including scavenging ROS and other free radicals, DNA synthesis, regulation of gene expression, and anti-inflammation. Many researchers have confirmed that Trx could resist oxidative stress, inhibit apoptosis, and promote cell proliferation, which makes it a protective enzyme against oxidative damage [32,33]. Identified from various organisms, three distinct forms of Trx have been characterized so far. Trx1 exists primarily in the cytosol and is also found in the nucleus and blood plasma. Trx2 is a mitochondrial form with an N-terminal mitochondrial translocation signal. The third isoform, SpTrx, is expressed exclusively in spermatozoa [34].

UVA and UVB in ultraviolet light can penetrate the seawater to a depth of at least 30 m, and even reach 60~70 m in Antarctic waters. Some marine plankton organisms have formed self-protection mechanisms to resist high levels of ultraviolet radiation on the sea surface [35,36]. In a previous study, we identified a novel Trx1 from the jellyfish *Cyanea capillata*, named CcTrx1. This is the first characterized Trx from a marine cnidarian. CcTrx1 exhibited high antioxidant activities and significantly protected cells against oxidative stress. As a marine antioxidant enzyme, CcTrx1 has significant advantages: it is highly active, abundant, and easy to obtain, and it possesses excellent prospects in antioxidant drug development. However, due to the natural barrier effect of cell membranes, delivery of macromolecular drugs to the cytosol remains a great challenge.

Currently, there are several commonly used methods to deliver protein and other biomacromolecules into the cells. Although microinjection, electroporation, liposome encapsulation, and viral transformation have been widely used in biomacromolecules’ transport, there are still many potential issues, such as triggering immune responses, high cytotoxicity as well as the unclear intracellular release efficiency of proteins [37,38,39,40]. A variety of bacterial exotoxins autonomously translocate to the cytosol to fulfill their function. Therefore, the use of bacterial toxins as shuttles provides new ideas for drug delivery systems, and various macromolecules have been successfully delivered into the cells via toxin-based vehicles in recent years [41,42]. Furthermore, toxin routes present paths into the cell that potentially cause less damage to the membrane and can be specifically targeted, contributing greatly to the development of drug delivery in the fields of cancer therapy, antivirals, and protein misfolding diseases. However, this strategy retains some limitations, including the limited selection of molecules that must fulfill multiple criteria and the existing immunity in human populations, which has to be assessed individually [43].

In comparison, protein modification by PTD has attracted increasing attention due to its high efficiency of membrane penetration, low cytotoxicity, and the ability to carry various biomacromolecules into the cells [21,22]. In recent years, CPPs have been mainly used in tumor-targeted therapy and drug delivery research, which enable drugs to enter into tumor cells and even penetrate epithelial cells and the blood-brain barrier [44,45]. Moreover, CPPs are particularly suitable for being fused with peptides or proteins with known sequences, and sufficient fusion proteins can be obtained efficiently by molecular cloning.

However, CPPs are rarely adopted in the exploitation of marine biological resources, except in breeding and vaccine delivery [46,47]. In this study, we innovatively combined TAT-PTD with CcTrx1 by molecular cloning and obtained sufficient amounts of PTD-CcTrx1 fusion protein with high purity. Our study clarified that PTD-CcTrx1 could exhibit specific transmembrane ability and still retain considerable antioxidant activities. PTD-CcTrx1 is the first reported fusion protein that binds marine antioxidant enzyme with PTD, and the method of using PTD to guide protein into the cells provides an important direction for the development and utilization of other marine active substances.

Exposure to H_2_O_2_ is used as a common model to observe the oxidative stress susceptibility of skin cells [48,49]. To explore the potential of PTD-CcTrx1 in protecting skin cells against oxidative damage, we established the H_2_O_2_-induced oxidative damage model in HaCaT cells and examined the inhibitory effects of PTD-CcTrx1 on H_2_O_2_-induced oxidative stress. The results indicated that PTD-CcTrx1 significantly inhibited the H_2_O_2_-induced intracellular oxidative stress and subsequently reversed the cytotoxicity and apoptosis. Several studies have reported that H_2_O_2_ could induce apoptosis by upregulating Bax/Bcl-2 ratio, promoting cytochrome c migration, and activating caspase 9 and caspase 3 [50,51]. In the present study, we found that molecular changes induced by H_2_O_2_ were significantly attenuated by the pretreatment of PTD-CcTrx1, demonstrating that PTD-CcTrx1 might inhibit apoptosis through the mitochondrial apoptosis pathway (Figure 8).

In addition, according to previous studies [32,52], Trx1 may directly scavenge ROS or donate hydrogen atoms to keep the levels of intracellular antioxidative substances such as GSH and NADPH, and subsequently stabilize the mitochondrial membrane to inhibit apoptosis. Here, we confirmed that the intracellular ROS levels were significantly reduced by PTD-CcTrx1. However, it is still necessary to explore the underlying mechanisms by which it reduced intracellular ROS levels, and thereby inhibited apoptosis and death caused by oxidative stress. Furthermore, other mechanisms of the anti-apoptotic effects of PTD-CcTrx1, such as binding to certain pro-apoptotic molecules, still need to be explored. This study provides a scientific basis for the feasibility of marine antioxidant enzymes entering cells to play biological roles and for further exploring their application in antioxidant drug development and UV radiation damage resistance.

## 4. Materials and Methods

### 4.1. Molecular Cloning

To construct the PTD-CcTrx1 expression vector, the coding gene of PTD-CcTrx1 incorporating NdeⅠ and XhoⅠ restriction enzyme sites was synthesized by Shanghai Generay Biotech Co., Ltd (Shanghai, China) and inserted into pET-24a plasmid. As shown in Appendix A, the double-digested PTD-CcTrx1 coding gene and pET-24a vector (Novagen, Madison, WI, USA) were ligated at 37 °C for 1 h using T4 DNA ligase (NEB, Ipswich, MA, USA), and then the ligation mixtures were transformed into *E. coli* TOP 10 competent cells (BioMed, Beijing, China). Successfully transformed cells were picked out using agar plates containing 100 μg/mL kanamycin, followed by nucleotide sequencing of both strands to confirm in-frame insertion through outsourcing service provided by Beijing Liuhe Huada Genomics Technology Co., Ltd. (Beijing, China). A similar procedure was used to construct the CcTrx1 expression plasmid without PTD as the negative control for the subsequent assays.

### 4.2. Expression and Purification of PTD-CcTrx1

Recombinant PTD-CcTrx1/pET-24a plasmids were transformed into *E. coli* BL21 for expression. Bacteria were cultured in LB medium with 50 μg/mL kanamycin at 37 °C with shaking for 4 h until the OD_600_ value reached 0.6–0.8. Expression of PTD-CcTrx1 fusion protein was induced by adding 0.2 mM isopropyl-β-D-thiogalactoside (IPTG) and shaking for 18 h at 37 °C. Cell pellets were collected by centrifugation (10,000× *g* for 8 min at 4 °C) and resuspended in the binding buffer (20 mM NaH_2_PO_4_, 500 mM NaCl, 30 mM imidazole, pH 7.4). Subsequently, the resuspended bacterial pellets were sonicated for 8 min in an ice bath and the lysates were centrifuged at 12,000× *g* for 30 min at 4 °C. The supernatants containing PTD-CcTrx1 were collected and purified by the ÄKTA purifier system using a 5 mL HisTrap HP Chelating column (GE Healthcare, Chicago, IL, USA). After the column was washed with 100 mL of the binding buffer (20 mM NaH_2_PO_4_, 500 mM NaCl, 30 mM imidazole, pH 7.4), the His-tagged PTD-CcTrx1 was eluted from the column by the addition of elution buffer (20 mM NaH_2_PO_4_, 500 mM NaCl, 500 mM imidazole, pH 7.4). A similar procedure was used to obtain the CcTrx1 without PTD as the control for the subsequent assays.

### 4.3. Immunoblotting

PTD-CcTrx1 was identified by immunoblotting. After electrophoresis on 12% SDS polyacrylamide gel, proteins were transferred to PVDF membrane and blocked with Western blot blocking liquid (Beyotime, Shanghai, China) for 1 h at room temperature. Transferred proteins were treated with the anti-His-tag antibody from mice (1:2000 dilution, Tiangen, Beijing, China) overnight at 4 °C, and then incubated with the HRP-labeled goat anti-mouse IgG (Beyotime, Shanghai, China) for 1 h at room temperature. Afterward, the membranes were washed for 30 min and then visualized using a chemiluminescent detection system (Syngene G: Box, Cambridge, UK).

### 4.4. Transmembrane Ability of PTD-CcTrx1

HaCaT cells were cultured in GIBCO Dulbecco’s Modified Eagle Medium: supplemented with 10% fetal bovine serum and 1% antibiotics (100 U/mL penicillin and 100 U/mL streptomycin) at 37 °C with 5% CO_2_ in a humidified atmosphere.

#### 4.4.1. Western Blot Assay

HaCaT cells were seeded in a 6-well plate for 24 h, and then incubated with 4 μM PTD-CcTrx1 for 15 min, 30 min, 1 h, 2 h. PBS and 4 μM CcTrx1 were used as a blank and negative control, respectively. Cells were rinsed with ice-cold PBS. A total of 80 μL of cell lysis buffer (Beyotime, Shanghai, China) was added to each well and shaken at 4 °C for 30 min, and then the cell lysates were scraped thoroughly and collected into centrifuge tubes. After centrifugation at 4 °C, 12,000× *g* for 15 min, the supernatants were collected for the Western blot assay. The mouse monoclonal antibody that could bind to the His tag of PTD-CcTrx1 was used as the primary antibody, and the secondary antibody was a goat anti-mouse HRP (horseradish peroxidase) labeled antibody.

#### 4.4.2. Immunofluorescence Assay

HaCaT cells were seeded in a 96-well plate for 24 h and then incubated with 4 μM PTD-CcTrx1 for 15 min, 30 min, 1 h, 2 h, with PBS and 4 μM CcTrx1 used as a blank and negative control. After being washed with ice-cold PBS, cells were fixed with 4% paraformaldehyde for 30 min, followed by permeabilization with Triton X-100 for 10 min at room temperature. After blocking with 3% BSA (bovine serum albumin) for 60 min, 100 μL of mouse anti-His-tag IgG (1:200 dilution) was added to each well at 4 °C overnight and then incubated with 100 μL of goat anti-mouse IgG/FITC (1:500 dilution) for 60 min at room temperature in the dark. Finally, cells were observed under a fluorescence microscope (Olympus CKX53, Tokyo, Japan), and the fluorescence intensity value was calculated and analyzed using ImageJ software (Version 1.8.0).

### 4.5. Antioxidant Capacity Assays in Cell-Free System

#### 4.5.1. Insulin Disulfide Reduction Assay by PTD-CcTrx1

Trx1 can catalyze the reduction of the two inter-chain disulfide bonds of insulin in the presence of DTT. A white precipitate is formed mainly from the free B chain of insulin, which is insoluble and can be monitored by measuring absorbance at 650 nm [53,54]. Hence, a turbidimetric assay was developed to measure the disulfide-reducing activity of PTD-CcTrx1 by recording the rate of precipitation by the free insulin chain B. The reaction mixture contained a final volume of 1 mL with 100 mM phosphate buffered saline (PBS), 1.25 mg/mL bovine insulin (Sigma-Aldrich, Shanghai, China), 2 mM ethylene diamine tetraacetic acid (EDTA), and different concentrations of purified PTD-CcTrx1 (2.5, 5, and 10 μM). PBS and the purified CcTrx1 were used as a blank and positive control, respectively. The reaction was initiated by adding 2 mL of 1 M DTT and monitored by measuring absorbance at 650 nm once a minute.

#### 4.5.2. DNA Cleavage Assay in a Metal-Catalyzed Oxidation System

In order to assess the ability of PTD-CcTrx1 to protect supercoiled DNA from oxidative damage, a DNA cleavage assay was performed in a metal-catalyzed oxidation (MCO) system according to the method described previously with slight modifications [20,55]. Briefly, the reaction was conducted at 37 °C for 2 h in 50 μL of reaction systems containing 1 μg pET-24a supercoiled plasmid DNA, 35 μM FeCl_3_, 10 mM DTT, and increasing concentrations of PTD-CcTrx1 ranging from 1 to 6 μM in 50 mM 4-(2-hydroxyethyl)-1-piperazineethanesul phonic acid (HEPES) (pH 7.0). PBS and 6 μM CcTrx1 were used as a blank and positive control, respectively. DNA was stained with GoldenView (BioMed, Beijing, China) and evaluated by 1% (*w*/*v*) agarose gel electrophoresis. The rate of the nicked form (NF) of supercoiled DNA represented the degree of oxidative damage and was analyzed using Image J software.

### 4.6. Protective Effect of PTD-CcTrx1 against H_2_O_2_-Induced Cytotoxicity

HaCaT cells were seeded in a 96-well plate at 1 × 10^4^ cells/well and cultured overnight. The cells were treated with different concentrations of H_2_O_2_ (0, 200, 400, 600, 800, 1000, 1200 μM) for 4 h. Then, 10 μL of CCK-8 reagents was added to each well and incubated at 37 °C for 2 h, respectively. Absorbance at 450 nm was measured using a microplate reader (BioTek, Winooski, VT, USA). Cell viability was calculated using the following formula:Viability (%) = (OD_Treated_ − OD_Blank_/OD_Untreated_ − OD_Blank_) × 100%

A cytotoxicity assay of PTD-CcTrx1 was also performed. First, HaCaT cells were seeded in a 96-well plate at 1 × 10^4^ cells/well and cultured overnight. The cells were treated with different concentrations of PTD-CcTrx1 (0, 0.01, 0.1, 1, and 10 μM) for 12 h. Then, cell viability was detected and calculated using the CCK-8 assay as above.

HaCaT cells were seeded in a 96-well plate at 1 × 10^4^ cells/well and cultured overnight. In a pretreatment protocol to examine the protective effect of PTD-CcTrx1 against H_2_O_2_-induced cytotoxicity, the cells were preincubated with PBS, 10 μM CcTrx1, and 10 μM PTD-CcTrx1 for 2 h, respectively, rinsed with PBS, and then exposed to 400 μM H_2_O_2_ for 30 min. In a posttreatment protocol, HaCaT cells were exposed to 400 μM H_2_O_2_ for 30 min, rinsed with PBS, and then incubated with PBS, 10 μM PTD-CcTrx1 and 10 μM CcTrx1 for 2 h, respectively. Finally, cell viability was detected and calculated using the CCK-8 assay.

### 4.7. Protective Effect of PTD-CcTrx1 on H_2_O_2_-Induced Apoptosis

#### 4.7.1. Cell Apoptosis Assay

Briefly, HaCaT cells were seeded in a 6-well plate at 1 × 10^6^ cells/well and cultured to the density of 70–80%. In a pretreatment protocol to examine the protective effect of PTD-CcTrx1 on H_2_O_2_-induced apoptosis, the cells were preincubated with PBS, 10 μM CcTrx1, and 10 μM PTD-CcTrx1 for 2 h, respectively, rinsed with PBS, and then exposed to 400 μM H_2_O_2_ for 30 min. In a posttreatment protocol, HaCaT cells were exposed to 400 μM H_2_O_2_ for 30 min, rinsed with PBS, and then incubated with PBS, 10 μM PTD-CcTrx1, and 10 μM CcTrx1 for 2 h, respectively. Afterward, cells were washed with ice-cold PBS, gently collected into sterile centrifuge tubes, and resuspended in 200 μL of the binding buffer from an apoptosis analysis kit (Beyotime, Shanghai, China). Then, 5 μL of FITC-Annexin V and 5 μL of propidium iodide (PI) were added into each tube for 15 min of staining. The cells were subjected to flow cytometry analysis with a flow cytometer (ThermoFisher Scientific, Waltham, MA, USA).

#### 4.7.2. LDH Activity Assay

Treated with CcTrx1 and PTD-CcTrx1 before or after the exposure to H_2_O_2_, supernatants of HaCaT cells in each well were collected for centrifugation, and then 120 μL of the supernatants were added into a 96-well plate for LDH activity detection. A total of 60 μL of LDH detection fluid was added to each well and incubated at room temperature for 30 min in the dark. Absorbance at 490 nm was measured using a microplate reader.

### 4.8. Total Antioxidant Capacity, ROS Level and Lipid Peroxidation Assays

Total antioxidant capacity (T-AOC) was detected using T-AOC Assay Kit with a rapid ABTS method (Beyotime, Shanghai, China). ABTS can be oxidized to green ABTS^+^ under the action of appropriate oxidants, and the production of ABTS^+^ will be inhibited by antioxidants. Thus, T-AOC of the sample can be determined and calculated by measuring the absorbance of ABTS^+^ at 414 nm. First, HaCaT cells were incubated with PBS and different concentrations of PTD-CcTrx1 (0.1, 1, 10 μM) for 2 h. Then, cells were washed with ice-cold PBS and incubated with 80 μL of cell lysis buffer at 4 °C for 30 min. The cell lysates were scraped thoroughly and collected into centrifuge tubes. After centrifugation at 4 °C, 12,000× *g* for 15 min, the supernatants were collected to detect the T-AOC, and the relative T-AOC values for each sample were normalized by the control group.

Further, to observe the effect of PTD-CcTrx1 on T-AOC of HaCaT cells exposed to H_2_O_2_, the cells were treated with 10 μM CcTrx1 and 10 μM PTD-CcTrx1 before or after the exposure to 400 μM H_2_O_2_. The T-AOC of HaCaT cells was detected, and the relative T-AOC values for each sample were normalized by the control group.

After treatment with 10 μM CcTrx1 and 10 μM PTD-CcTrx1 before or after the exposure to 400 μM H_2_O_2_, HaCaT cells were collected into sterile centrifuge tubes, washed twice with PBS, and then stained with 2,7-dichlorodi-hydrofluorescein diacetate (DCFH-DA, Beyotime, Shanghai, China). The reaction between ROS and DCFH-DA resulted in dichlorofluorescein (DCF), a compound that emits green fluorescence, which can be detected under a fluorescence microplate reader (ThermoFisher Scientific, Waltham, MA, USA). The fluorescence value indicates the ROS level in cells.

For the lipid peroxidation assay, we used a commercial kit (Beyotime, Shanghai, China) to quantify the generation of malondialdehyde (MDA) according to the manufacturer’s protocol with slight modifications [56]. After the pre- or posttreatment protocol, HaCaT cells were washed with ice-cold PBS and treated with 80 μL of cell lysis buffer at 4 °C for 30 min. Cell lysates were scraped thoroughly and collected into centrifuge tubes. After centrifugation at 4 °C, 12,000× *g* for 15 min, the supernatants were collected to quantify the MDA level.

### 4.9. Effects of PTD-CcTrx1 on Apoptosis Signaling Pathways Activated by H_2_O_2_ in HaCaT Cells

HaCaT cells were treated with 10 μM CcTrx1 and 10 μM PTD-CcTrx1 before or after the exposure to 400 μM H_2_O_2_. The cells were washed with ice-cold PBS, and cytoplasmic and mitochondrial proteins were extracted according to the manufacturer’s instructions [57]. First, the cells were collected by centrifugation (1000× *g*, 4 min) and resuspended using mitochondria isolation reagents (Beyotime, Shanghai, China). After 10 min of incubation in an ice bath, the cell homogenates were centrifugated at 4 °C, 12,000× *g* for 10 min, and the pellets containing mitochondrial proteins were extracted from the supernatants containing cytoplasmic proteins. Both mitochondrial and cytoplasmic proteins were prepared for the subsequent Western blot assay. After SDS-PAGE and membrane transfer procedures, the membranes were incubated with primary antibodies overnight, including cleaved caspase 3 (1:1000, Beyotime, Shanghai, China), cleaved caspase 9 (1:1000, Beyotime, Shanghai, China), Bax (1:1000, Abcam, Cambridge, UK), and Bcl-2 (1:1000, Abcam, Cambridge, UK) for cytoplasmic protein analysis. The content of cytochrome c (1:2000, Beyotime, Shanghai, China) in mitochondria was also investigated. Meanwhile, β-Actin (1:2000 Beyotime, Shanghai, China) and COX-IV (1:5000, Abcam, Cambridge, UK) were used as the internal reference, respectively. After being incubated with the conjugated secondary antibodies, the bands were visualized using a chemiluminescent detection system and analyzed via Image J software.

### 4.10. Statistical Analysis

The experimental data were analyzed using GraphPad Prism 8 statistical software. The comparison of means between two groups was performed by t-test, and differences among three or more groups were analyzed using a one-way ANOVA test. A value of *p* < 0.05 was considered statistically significant. All data were represented as mean ± SD, and the experiments were performed in triplicates.

## 5. Conclusions

In conclusion, we obtained a novel recombinant protein PTD-CcTrx1 and applied it for intracellular oxidative damage protection for the first time. PTD-CcTrx1 retained specific transmembrane ability and antioxidant capacities. It also exhibited significant protective effects against H_2_O_2_-induced damage in HaCaT cells, which may be attributed to a blockage on apoptosis-related pathways. Our study promotes the application of PTD-CcTrx1 as a novel antioxidant to treat skin oxidative damage and provides a scientific basis for drug development of other novel marine-derived molecules in the future.

## Figures and Tables

**Figure 1 ijms-24-07340-f001:**
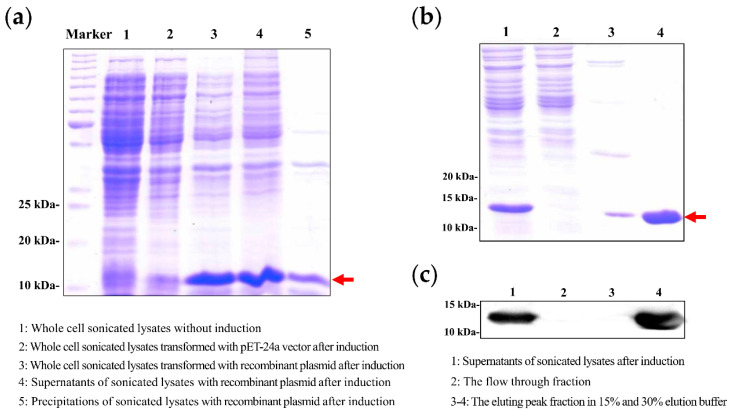
Expression, purification, and identification of PTD-CcTrx1. Recombinant PTD-CcTrx1 plasmids were transformed into *E. coli* BL21, and then induced by 0.2 mM IPTG with shaking for 18 h at 37 °C. Cell pellets were collected, resuspended, and sonicated, and then centrifugated at 12,000× *g* for 30 min at 4 °C. The supernatants containing PTD-CcTrx1 were purified via the ÄKTA purifier system using a 5 mL HisTrap HP Chelating column. (**a**) SDS-PAGE analysis of the samples under different expression conditions. (**b**) SDS-PAGE analysis of the samples collected from different steps of purification. (**c**) Western blot analysis of the samples collected from different steps of purification using the anti-His-tag antibody. The position corresponding to PTD-CcTrx1 is indicated by the red arrow.

**Figure 2 ijms-24-07340-f002:**
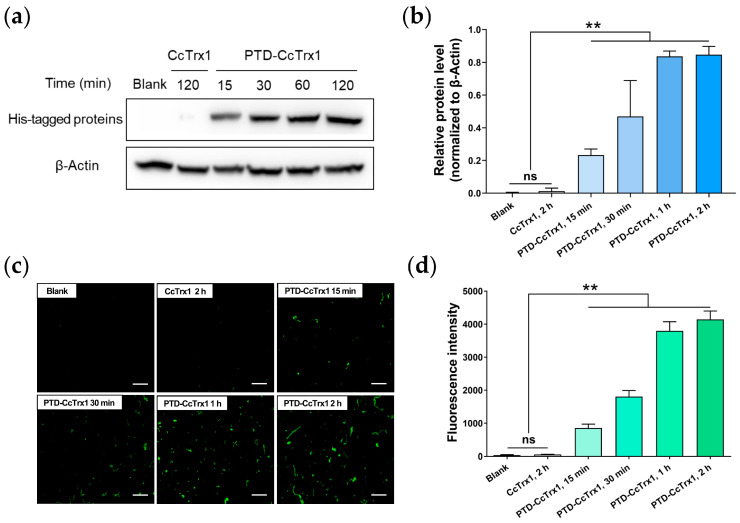
Transduction of PTD-CcTrx1 into HaCaT cells. First, HaCaT cells were incubated with 4 μM PTD-CcTrx1 for 15 min, 30 min, 1 h, 2 h, respectively. PBS and 4 μM CcTrx1 were used as a blank and negative control. Then, cells were rinsed with ice-cold PBS thoroughly, and the intracellular His-tagged proteins were detected by the anti-His-tag antibody via the Western blot assay and immunofluorescence assay. (**a**) Western blot analysis of intracellular His-tagged proteins using anti-His-tag antibody at different times. (**b**) Intracellular His-tagged proteins were quantified using densitometric analysis with β-Actin as the internal control. (**c**) Intracellular amount of His-tagged proteins at different times after incubation detected by immunofluorescence assay. Bar = 100 μm. (**d**) The fluorescence data were quantified by ImageJ software. Quantification data were obtained from three independent experiments, and the data are presented as means ± SD (*n* = 3) (ns, no significance, ** *p <* 0.01 vs. blank group).

**Figure 3 ijms-24-07340-f003:**
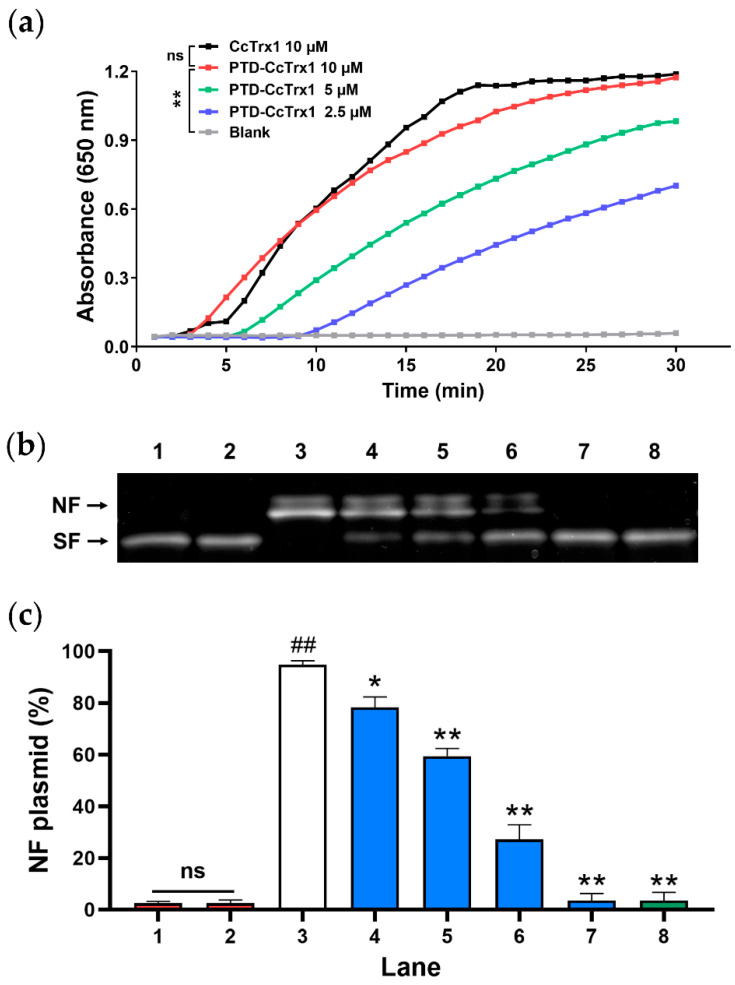
Antioxidant activities of PTD-CcTrx1 in a cell-free system. (**a**) Disulfide reductase activity of PTD-CcTrx1. Insulin disulfide reduction activities of different concentrations of PTD-CcTrx1 and 10 μM CcTrx1 were measured in the presence of 2 mM DTT (ns, no significance, ** *p* < 0.01). (**b**,**c**) Effect of PTD-CcTrx1 on protecting supercoiled plasmid DNA against oxidative damage. Lane 1: pET-24a plasmid DNA alone; lane 2: pET-24a plasmid DNA, and 10 mM DTT; lane 3: pET-24a plasmid DNA, 10 mM DTT, and 35 μM FeCl_3_; lanes 4–7: pET-24a plasmid DNA, 10 mM DTT, 35 μM FeCl_3_, and different concentrations of PTD-CcTrx1 (1, 2, 4, 6 μM, respectively); lane 8: pET-24a plasmid DNA, 10 mM DTT, 35 μM FeCl_3_, and 6 μM CcTrx1. The bands corresponding to the nicked form (NF) and supercoiled form (SF) are indicated on the left side. Quantification data were obtained from three independent experiments, and the data are presented as means ± SD (*n* = 3) (ns, no significance, ## *p* < 0.01 vs. lane 1; * *p* < 0.05, ** *p* < 0.01 vs. lane 3).

**Figure 4 ijms-24-07340-f004:**
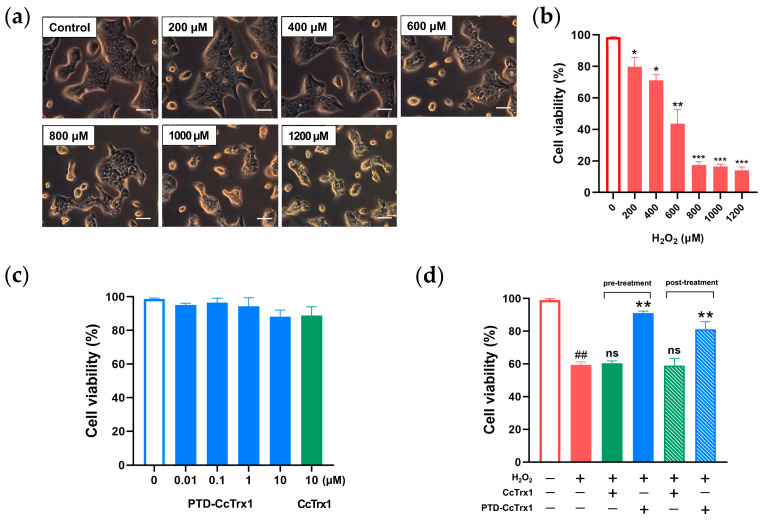
Protective effects of PTD-CcTrx1 against H_2_O_2_-induced cytotoxicity in HaCaT cells. (**a**) Representative morphological changes of HaCaT cells, which were treated with different concentrations of H_2_O_2_ and observed with a phase contrast microscope. Bar = 100 μm. (**b**) HaCaT cells were treated with increasing concentrations of H_2_O_2_ (0, 200, 400, 600, 800, 1000, 1200 μM) for 4 h (* *p* < 0.05, ** *p* < 0.01, *** *p* < 0.001 vs. control group). (**c**) HaCaT cells were treated with increasing concentrations of PTD-CcTrx1 (0, 0.01, 0.1, 1, 10 μM) and 10 μM CcTrx1 for 12 h. (**d**) HaCaT cells were treated with 10 μM PTD-CcTrx1 and 10 μM CcTrx1 before or after the exposure to 400 μM H_2_O_2_. All treatments were performed in triplicate individually, and the data are presented as means ± SD (*n* = 3) (## *p* < 0.01 vs. control group; ns, no significance, ** *p* < 0.01 vs. H_2_O_2_ alone group).

**Figure 5 ijms-24-07340-f005:**
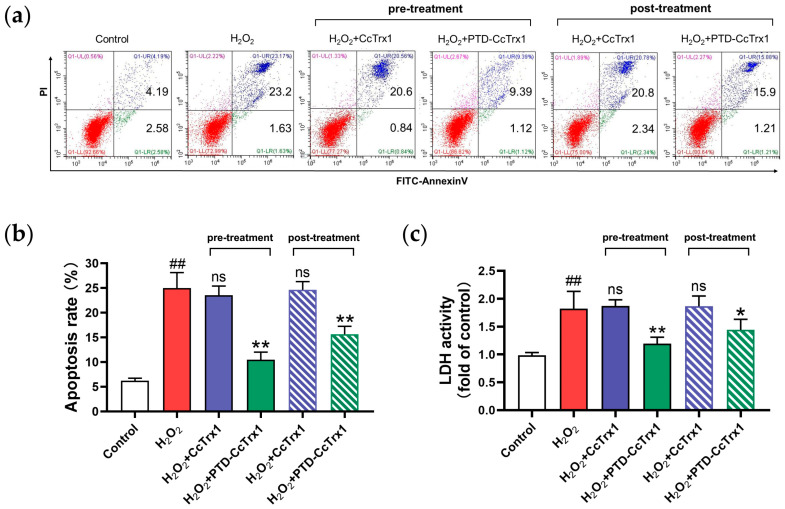
Protective effects of PTD-CcTrx1 against H_2_O_2_-induced apoptosis and necrosis in HaCaT cells. First, HaCaT cells were treated with 10 μM PTD-CcTrx1 and 10 μM CcTrx1 before or after the exposure to 400 μM H_2_O_2_. After treatment, the cells were resuspended in the binding buffer of the apoptosis analysis kit and stained with Annexin V-FITC/PI, and then analyzed by flow cytometry. (**a**) PTD-CcTrx1 significantly inhibited H_2_O_2_-induced apoptosis of HaCaT cells. (**b**) Histogram of the apoptosis rate in different groups. (**c**) LDH activity of cell supernatants in different groups. All treatments were performed in triplicate individually, with values expressed as mean ± SD (## *p* < 0.01 vs. control; ns, no significance, * *p* < 0.05, ** *p* < 0.01 vs. H_2_O_2_ alone group).

**Figure 6 ijms-24-07340-f006:**
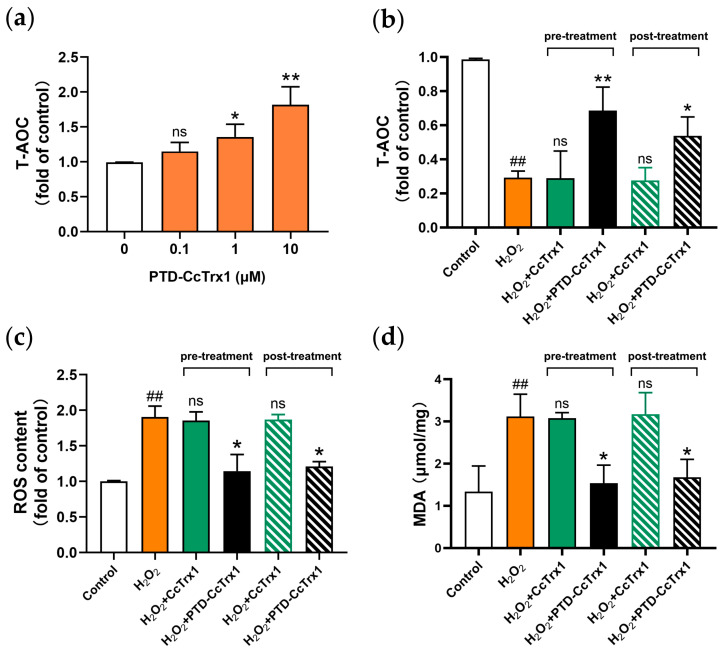
Attenuation of H_2_O_2_-mediated oxidative stress in HaCaT cells by PTD-CcTrx1. (**a**) The total antioxidant capacity (T-AOC) in HaCaT cells after treatment with different concentrations of PTD-CcTrx1 for 2 h (ns, no significance, * *p* < 0.05, ** *p* < 0.01 vs. control group). (**b**) PTD-CcTrx1 attenuated the decrease of T-AOC induced by H_2_O_2_. HaCaT cells were treated with 10 μM CcTrx1 and 10 μM PTD-CcTrx1 before or after the exposure to 400 μM H_2_O_2_. Afterward, the cells were rinsed with ice-cold PBS and treated with 80 μL of cell lysis buffer. Then, cell lysates were scraped thoroughly for centrifugation, and the supernatants were collected to detect the T-AOC. (**c**) ROS increase induced by H_2_O_2_ was attenuated by PTD-CcTrx1. After treatment, HaCaT cells were rinsed with PBS before undergoing DCFH-DA staining and then detected under a fluorescence microplate reader. (**d**) Analysis of MDA generation and lipid peroxidation. After treatment, cell lysates were scraped thoroughly and the supernatants were collected to quantify the MDA level. All treatments were performed in triplicate individually, with values expressed as mean ± SD (## *p* < 0.01 vs. control; ns, no significance, * *p* < 0.05, ** *p* < 0.01 vs. H_2_O_2_ alone group).

**Figure 7 ijms-24-07340-f007:**
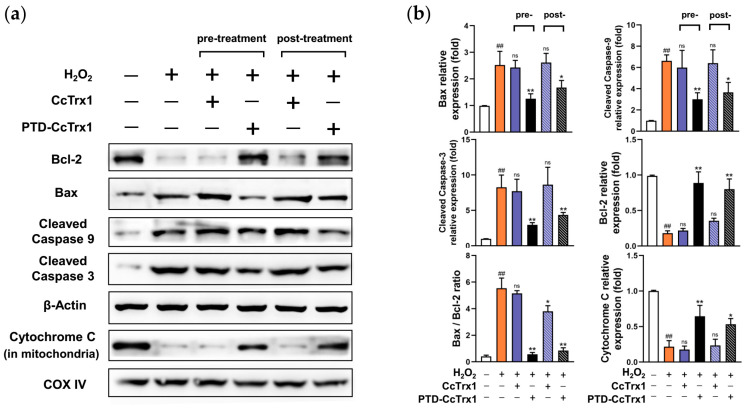
PTD-CcTrx1 inhibited the intrinsic apoptotic pathway in H_2_O_2_-treated HaCaT cells. First, HaCaT cells were treated with 10 μM CcTrx1 and 10 μM PTD-CcTrx1 before or after the exposure to 400 μM H_2_O_2_. The cells were washed with ice-cold PBS, and cytoplasmic and mitochondrial proteins were extracted for the Western blot assay. (**a**) Western blot analysis of Bax, Bcl-2, cleaved-caspase 3, cleaved-caspase 9, and cytochrome c (in mitochondria) of each group. β-Actin and COX IV were set as internal references, respectively. (**b**) Relative contents of the proteins were calibrated by setting the control group as one. All treatments were performed in triplicate individually, and the data are presented as means ± SD (*n* = 3) (## *p <* 0.01 vs. control; ns, no significance, * *p <* 0.05, ** *p <* 0.01 vs. H_2_O_2_ alone group).

**Figure 8 ijms-24-07340-f008:**
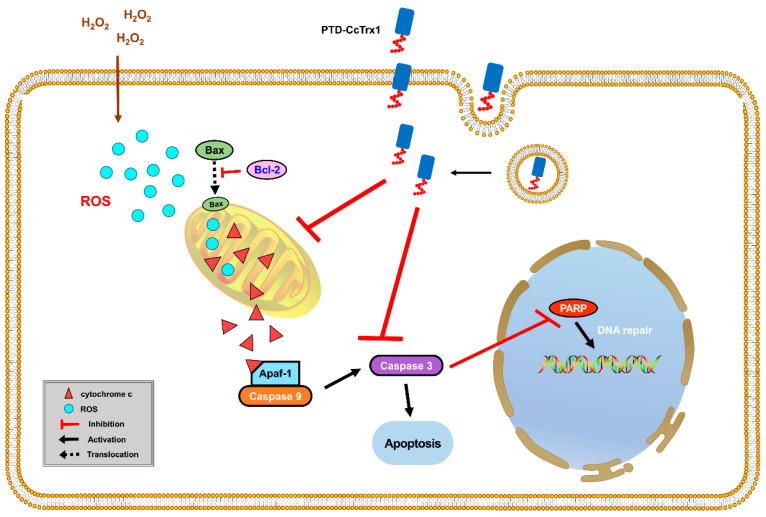
Possible mechanisms of the protective effect of PTD-CcTrx1 against H_2_O_2_-induced apoptosis in HaCaT cells. H_2_O_2_ induced the accumulation of intracellular ROS, which induced apoptosis by promoting the translocation of Bax and cytochrome c and then activating caspase 9 and caspase 3. PTD-CcTrx1 could penetrate the cell membrane and protect against H_2_O_2_-induced apoptosis by downregulating the Bax/Bcl-2 ratio, inhibiting the translocation of cytochrome c, and suppressing the activation of caspase 9 and caspase 3.

## Data Availability

The data used to support the findings of this study are available from the corresponding author upon request.

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
