# Peer review of "Protective Effects of a Jellyfish-Derived Thioredoxin Fused with Cell-Penetrating Peptide TAT-PTD on H2O2-Induced Oxidative Damage"

_ijms, 2023, doi:10.3390/ijms24087340_

Round 1
Reviewer 1 Report
In this manuscript, the authors obtained a recombinant protein PTD-CcTrx1, which is the first reported fusion protein of marine antioxidant enzyme and protein transduction domain (PTD). They found that PTD-CcTrx1 exhibited specific transmembrane ability and antioxidant activities, significantly attenuated the intracellular oxidative stress, and protected HaCaT cells from oxidative damage. This work is interesting and well done. A detailed analysis has been accomplished in the manuscript. However, minor revisions are needed to improve this manuscript.
1. Materials and Methods: 4.2. Expression and Purification of PTD-CcTrx1. The authors expressed the PTD-CcTrx1 under conditions of an IPTG concentration of 0.2 mM, an induction time of 18 h, and an induction temperature of 37℃. Did you choose these conditions according to any references? These conditions are very important for the subsequent experiments.
2. Results 2.2 and 2.3: Scale bars should be included in Figure 3c under the microscope. More experimental details need to be added in the legends of Figure 3 and Figure 2, to understand the results without referring to the main text.
3. Results 2.5: Statistical symbols (*, # or ns) should be included in Figure 5b if you have averaged the three separate experiments.
Reviewer 2 Report
I have reviewed the article Protective Effects of PTD-CcTrx1 on H2O2-induced Oxidative 2 Damage in HaCaT Cells, in which the authors report cloning the ORF of Trx1 from a marine organism, fusing the coding gene to a Protein Transduction Domain and expressing it in E. coli cells. They purified the protein and used it to pretreat HaCaT cells, demonstrating that PTD-CcTrx1 alone can mitigate the many damages caused by H2O2 treatment.
My only suggestion to the authors is to put the reference on the third reported isoform of Trx, found only in spermatozoa (lines 237-238).
Author Response
Thank you for your professional suggestion. We have added the reference in our revised manuscript listed as [34] (line 250).
Reviewer 3 Report
In this manuscript, the authors developed an anti-oxidation recombinant protein by fusing cell penetrating peptide with a newly identified jellyfish thioredoxin (PTD-CcTrx1). PTD-CcTrx1 can enter HaCaT cells and inhibit the H2O2-induced oxidative stress and apoptosis. The experiments were designed and performed well, the manuscript was written clearly and precisely. The findings are very interesting and could potentially cause sufficient attention to the technology-minded readership. In order to bring this manuscript to the publication level, couple of issues need to be addressed:
1. I suggest to modify the title by changing “PTD-CcTrx1” to a more precise description, to avoid the abbreviation and make the term more clear to the readers without much background on the previous studies.
2. Some of the figures, such as figure 1, 2, and 10, can be moved to a supplementary material to keep the main text less redundant - nowadays molecular cloning and protein purification are quite common and routine experiments in the labs.
3. For figure 3, based on the method section, “intracellular” is not accurate. Basically the authors cannot distinguish the molecules tightly attached on the cell membrane, either on the cell surface or partially embedded in the membrane, and the molecules really reach the cytosol. The right way to do this type of experiment should be in “pulse-chase” manner: (1) incubate your protein with cells at low temperature (such as on ice); (2) remove extra protein in the solution (PBS washing and changing medium); (3) incubate cells at 37 degree let your protein enter cells; (4) remove the extra surface protein (trypsin digestion); (5) finally analyze the protein inside of cells. Please repeat these experiments.
4. For figure 5-8, the authors always treat cells by PTD-CcTrx1 firstly (the extra Trx1 in the solution has been removed or not?) and H2O2 secondly. Although CcTrx1 without PTD can serve as a good control to demonstrate the inhibition effect is happened inside of cells, I assume as a potential therapeutics, the oxidative damage should be induced earlier then the anti-oxidative reagents arrive. The authors need to show at least one set of experiment by adding H2O2 before PTD-CcTrx1.
5. There are multiple recent studies used the similar protein fusing strategy but using bacterial toxins as vehicle to deliver cargoes into targeted cells (for example PMID: 35263584), which are showing good efficiency and better specificity than CPPs. These recent advances on protein delivery field are recommended to be discussed in the Discussion section.
Round 2
Reviewer 3 Report
The authors have answered all my concerns.